# Triple-Shape Memory Behavior of Modified Lactide/Glycolide Copolymers

**DOI:** 10.3390/polym12122984

**Published:** 2020-12-14

**Authors:** Anna Smola-Dmochowska, Natalia Śmigiel-Gac, Bożena Kaczmarczyk, Michał Sobota, Henryk Janeczek, Paulina Karpeta-Jarząbek, Janusz Kasperczyk, Piotr Dobrzyński

**Affiliations:** 1Centre of Polymer and Carbon Materials, Polish Academy of Sciences, M. Curie-Sklodowska 34, 41-819 Zabrze, Poland; ngac@cmpw-pan.edu.pl (N.Ś.-G.); bkaczmarczyk@cmpw-pan.edu.pl (B.K.); msobota@cmpw.edu.pl (M.S.); hjaneczek@cmpw-pan.edu.pl (H.J.); pkarpeta@cmpw-pan.edu.pl (P.K.-J.); jkasperczyk@sum.edu.pl (J.K.); 2School of Pharmacy with the Division of Laboratory Medicine in Sosnowiec, Medical University of Silesia, 40-000 Katowice, Poland; 3Faculty of Science and Technology, Jan Dlugosz University, 42-200 Czestochowa, Poland

**Keywords:** blends, shape memory, polylactide, esters, bioresorbable polyesters

## Abstract

The paper presents the formation and properties of biodegradable thermoplastic blends with triple-shape memory behavior, which were obtained by the blending and extrusion of poly(l-lactide-*co*-glycolide) and bioresorbable aliphatic oligoesters with side hydroxyl groups: oligo (butylene succinate-*co*-butylene citrate) and oligo(butylene citrate). Addition of the oligoesters to poly (l-lactide-*co*-glycolide) reduces the glass transition temperature (T_g_) and also increases the flexibility and shape memory behavior of the final blends. Among the tested blends, materials containing less than 20 wt % of oligo (butylene succinate-*co*-butylene citrate) seem especially promising for biomedical applications as materials for manufacturing bioresorbable implants with high flexibility and relatively good mechanical properties. These blends show compatibility, exhibiting one glass transition temperature and macroscopically uniform physical properties.

## 1. Introduction

Materials displaying shape memory properties offers the possibility to recall the material’s predefined permanent shape from a temporary intermediate shape in response to numerous stimuli such as temperature increase, magnetic or electric field, radiation, and pH changes [1,2]. Shape memory in polymeric materials occurs as a result of the appropriate construction of this material being a combination of chemical composition and chains microstructure. The properly selected technological process of creating and programming a permanent shape and generating a temporary shape by mechanical deformation is equally important. The most common stimulus causing spontaneous shape transformations is supplied heat. This type of thermo-responsive polymer is discussed in this paper. All causes of the thermally induced shape memory effect of polymers were divided into three mechanisms [3]. The first dual state mechanism is a commonly observed phenomenon in most polymers. The proceeding shape transitions is related to the presence of glass transition (from glass to rubbery state) at the polymer glass transition temperature. However, an efficient shape memory effect requires a specific polymer structure divided into the presence of elastic elements, so called-switches, allowing the mechanical deformation of the material and simultaneous occurrence of nodes-rigid structures, most often arising by binding polymer chains together by chemical covalent bonds or physical intermolecular interactions [4,5]. The second mechanism, called dual-component, is typical for the polymers that have clearly separated hard and soft segments as well for polymeric blends or hybrid polymers with an inclusion-matrix type of microstructure. The third mechanism, the partial transition mechanism, is rather rare; it is related to the ability to carry out the special shape programing in blends contained low melting dispersed phase or segmented and block polymers [3].

The classic shape memory polymers possess the ability to transit between the two shapes: one permanent shape requiring programming and the other temporary shape. Based on the posted literature reports, polymeric materials can remember three (triple-shape memory polymers, TSMPs) and four or even five shapes (multiple-shape memory polymers, MSMPs). A process for the creating multiple-shape memory effect comprises more than one step. In each step, at a different temperature, only one temporary shape is fixed. The shape memory effect of several shapes is explained by the occurrence of a few glass transition temperatures as well the occurrence of two or more separate molecular switches related to the structure of the polymer chain (discrete thermal transitions, two independent switch units associated with two different glass transitions temperatures, molecular switches by covalent bonds, and supramolecular interactions either by themselves or in combination with a phase transition switch) [6,7,8,9]. L. Sun and W. Huang presented the mechanism of creating the shape memory effect, which could serve as a starting point for optimized synthesis and shape programming to achieve an enhanced performance of the phenomenon of polymers multi-shape memory effect [10]. In the proposed models, the authors confirm that the cause of this effect is not limited to only the presence of the polymer glass transition temperature. An example of TSMPs with tuneable intermediate shapes by glass transition temperatures is the miscible blend poly (l-lactide)/poly (methyl methacrylate) obtained by Samuel et al. [8]. A close correlation of the shape change temperature with the temperature of the shape programming was shown. MSMPs with a broad transition temperature were obtained by Wang et al. [9] as a result of the synthesis of methacrylate-based polymers bearing cholic acid and oligo (ethylene glycol) pendants. For described copolymers, the glass transition temperatures and the range of transition temperatures at which the shape changes, respectively, can be adjusted by varying the proportion of monomers. The degree of recovery ratios of these copolymers is high, above 83–99%. The phenomenon of occurrence of two well-separated glass transition temperatures resulting in several-shape memory in polymers has been presented by the Gu et al. [11] for composites obtained through blending segmented polyurethane and poly(l-lactide)-*block*-poly(tetramethylene) copolymer. The results show that the blends can memorize two temporary shapes in a single shape memory cycle, indicating the blends’ triple-shape memory effects. Compared with multiple-shape memory with only one broad reversible phase transition, the current triple-shape memory polymer system possesses two well-separated phase structures and has two separated glass transitions at around −30 and 46 °C, respectively. Voit et al. have demonstrated the TSMPs in which the effect of several-shape memory was based on a combination of glass transition temperature and the dissociation of self-complementary hydrogen bonding moieties in polymers obtained of methacrylated 2-ureido-4 [1H] pyrimidone copolymerized with desired ratios of n-alkyl acrylates and a bisphenol A-based diacrylate cross-linker [7]. With increasing acrylate monomer concentration, the onset of recovery occurs at higher temperatures and the separation of the two-recovery occurrence decreases.

The other relatively simple way permits obtaining a polymeric material that shows a quadruple-shape memory behavior was presented by H. Li et al. [12]. This material, with well-distributed multiple nanophases was formed by blending a series of block styrene/methacrylate copolymers in the form of nano-latexes. The formed nanophases act as transition phases with variable transition temperatures, which can be adjusting by the St/MA ratio and are the cause of the observed phenomenon of multiple-shape memory. Similar material was obtained by a series of triple-shape memory polymer materials composed of paraffin and lightly cross-linked by vulcanization *trans*-polyisoprene (TPI) [13] As shown in the published literature, although the presence a chain cross-link is not necessary to obtain the shape memory effect, the simultaneous occurrence of both physical and chemical cross-links phenomena allows obtaining polymeric materials with especially high strain and multiple-shape memory [14].

Today, one of the most important applications of shape memory polymers is the formation of special devices and materials for medicine and veterinary medicine, including self-tightening sutures [15], self-expanding stents [16], endovascular clot removal [17], and tridimensional scaffold for tissue engineering [18], as well drug release systems [19]. For these applications, polymers must necessarily meet many conditions, such as biocompatibility, sterilizable, easy processing, and good mechanical properties, and the switching temperature should be close or slightly above body temperatures. Additionally, the thermoplastic bioresorbable shape memory polymers appear to be particularly valuable for biomedical applications. Using such tools produced by thermoplastic processing or 3D printing as self-expanding stents, self-clamping clamps, and loops, as well self-expanded scaffolds for the treatment of large bone defects, can introduce or improve many innovative surgical techniques. The most common polymers used for these purposes are aliphatic polyesters or polyester–carbonates. A classic example is the homopolymer poly (l-lactide) (PLLA) or the copolymer l-lactide-*co*-glycolide (PLLAGL), today commonly used in surgery i.e., materials with a well-known biocompatibility and biodegradability, with good mechanical properties. The main disadvantage of the shape memory products based on PLLA and PLLAGL is the relatively high temperature required to initiate the process of returning to the permanent shape, which limits the use of this material in medical application [12,20]. It is known that programing the temporary shape of the products formed with PLLA at temperature close to glass transition by high mechanical deformation is practically impossible without destroying the sample. In this case, the highly deformed temporary shape can only be formed at temperatures higher than the glass transition temperature. The process of the return to permanent shape proceeds with a relatively low recovery stress. [21]. In order to improve the PLLA’s ability to plastic deformation, blending with other polymers and plasticizers is used. It is a relatively simple and cheap method of modifying polymeric materials, enabling the adjustment or improvement of mechanical properties and shape memory parameters, such as the temperature of change in the shape of conventional SMP polymers [22]. Lowering the programming temperature of a temporary shape of the PLLA by adding polyurethane has been shown, e.g., Lai et al. [23]. He obtained shape memory material by blending PLLA with thermoplastic polyurethane (TPU, Desmopan) in a ratio of 70/30 and 50/50. The addition of polyurethane to the polylactide improved the flexibility of the material and allowed deformation at 25 °C. In the case of a blend with an equilibrium composition, the addition of TPU significantly increased the degree of shape recovery. Blending PLLA with low molecular weight poly (ethylene glycol) improves the elongation at break and softness [24,25]. An interesting composition with shape memory and switching temperature close to the body temperature was obtained, also blended PLLA with low molecular oligomeric PLA and chitosan. The enhancement of the blend shape memory and antibacterial properties was obtained by the addition of silver nanoparticles [26]. Polycaprolactone (PCL) is an alternative, well-known, FDA-approved and also biodegradable polyester. The PCL chain is softer and more flexible than the PLA and with very low glass transition temperature. Due to this, the polymers’ biomedical applicability, containing caproil units and shape memory behavior, are really only ε-caprolactone copolymers or cross-linked polycaprolactone [27]. The comparable properties are obtained also by blending. For example, polycaprolactone (PCL) mixed with ethylene vinyl-acetate copolymer (EVA) form the multiple-shape memory cross-linked blends [28].

In the previous published studies, due to the large differences in physical properties, chain elasticity, and the course of biodegradation, very interesting physicochemical properties of block copolymers (poly (l-lactide)-*block*-poly(succinate succinate)) or blends of polylactide with poly (butylene succinate) (PBS) seem particularly useful to use in the controlled release of drugs and the formation of bioresorbable implants with high elasticity. In the published studies, very interesting physicochemical properties of block copolymers poly(l-lactide)-*block*-poly (butylene succinate) or blends of polylactide with polybutylene succinate have been demonstrated [29,30,31]. Aliphatic polyesters consist of chain fragments with high biocompatibility, such as citric or succinic acid derivatives related to their role in the metabolic cycle of humans. PBS is similar to the materials most popular in the biomedical applications: polylactides or copolymers of lactide with glycolide, aliphatic polyester, which are fully biodegradable and biocompatible [32,33,34].

In this study, various aspects of the thermal, mechanical, and shape memory properties of the PLLAGL blends with different amounts of oligo (butylene succinate), oligo (butylene succinate-*co*-butylene citrate), and oligo (butylene citrate) were investigated. Obtained materials exhibit interesting physical–chemical properties, particularly in applications for the controlled release of drugs, or the formation of bioresorbable implants with high flexibility and shape memory behavior.

## 2. Materials and Methods

### 2.1. Materials

Zirconium (IV) acetylacetonate (Sigma Aldrich, Saint Louis, MO, USA), 1-4-butanediol (Sigma Aldrich, Saint Louis, MO, USA), methyl alcohol (Avantor, Gliwice, Poland), chloroform (Avantor, Gliwice, Poland), dimethyl succinate 98% (Sigma Aldrich, Saint Louis, MO, USA), triethyl citrate 98% (Sigma Aldrich), and titanium (IV) butoxide 97% (Sigma Aldrich, Saint Louis, MO, USA) were applied in commercial form. l-Lactide and glycolide (Glaco Ltd., Beijin, China) were purified by recrystallization from ethyl acetate solution and dried in a vacuum oven at room temperature.

#### Copolymers and Blends Preparation 

The copolymerization of poly(l-Lactide-co-glycolide) (PLLAGA) was conducted in bulk at 120 °C with Zr(acac)_4_ as initiator (initiator/monomer ratio 1:1200) according to the method described previously [35]. The composition of the PLLAGA was 85% mol. of l-lactidyl units and 15% mol. glycolidyl units, the molecular weight was 134,000 g/mol. and the mass dispersion was about 2.

The oligo (butylene succinate) (BS) was synthesized by the transesterification of succinic acid methyl di-ester with 1,4-butanediol, according to the method described detailed previously [36]. This copolymer presents the linear microstructure of the chain. The oligo (butylene citrate) (CA) and oligo (butylene succinate-*co*-butylene citrate) (BSCA) with a 40/60 molar ratio of derivatives of citric acid to succinic acid were synthesized by the analogous method. In turn, these oligomers, due to the presence of citric acid derivatives, presented a branched chain microstructure with side hydroxyl groups [36]. The chain structures of used oligoesters are pictured in Scheme 1.

Polymer blends were obtained by the extrusion of a previously prepared PLLAGA copolymer mixture and an appropriate amount of the selected oligomer. The extrusion process was carried out at a temperature of about 180 °C using a Haake Minilab II extruder. The product was obtained in the form of wire and pellets.

Based on earlier preliminary tests, aliphatic oligoesters with an approximate mass of 4000–6000 g/mol. were optimal; therefore, the polycondensation reaction was discontinued when the molecular weight of the synthesized oligomers was about 5000 g/mol. The basic properties of the obtained oligomers are shown in Table 1.

### 2.2. Characterizations

The samples selected for the shape-memory and tensile tests had the dog-bone shape with a total length of 110 mm, gauge length of 35 mm, width of 5 mm, and thickness of 1.5 mm. These samples were formed with an injection molding machine Haake MiniJet II Thermo Fisher Scientific, Waltham, MA, USA). The process of obtaining the samples was carried out at a temperature about 180 °C, using a previously obtained blend in the form of pellets. The tensile test was carried out according to ISO 527:1998 standard at ambient temperature (25 °C) or at the temperature of the body (37 °C). The test was performed with an Instron 4200 tensile testing machine equipped with a temperature controlled environmental chamber and a mechanical extensometer. The strain rate was about 0.01 s^−1^ in all of the experiments. An average value obtained for 4–5 specimens was employed. The standard deviation was calculated, too.

Thermal analyses of formed materials were specified with using differential scanning calorimetry (DSC apparatus Du Pont1090B, Du Pont Instruments Corp., COLUBUS, OH, USA), the calibration of gallium and indium, with a heating rate of 20 °C/min-according to the ASTM E 1356-08 standard. Infrared spectra FTIR were recorded on a JASCO FT/IR-6700 spectrometer (Jasco, Tokyo, Japan) with a resolution of 2 cm^−1^ as a result of the accumulation of 64 scans. Samples in solid state were analyzed in situ using the ATR method. The second derivative spectra were calculated using the Savitzky–Golay method, degree of polynomial: 2, number of convolution points: 20. Morphology of blends was observed by scanning electron microscopy (SEM) (Quanta 250 FEG, FEI Company, Hillsboro, OR, USA). SEM was operating under low vacuum conditions (80 Pa) and an acceleration voltage of 5 kV from secondary electrons collected by a Large Field Detector. The dual shape-memory property of blends was assessed with an Instron 4200 tensile testing machine equipped with a thermostated chamber and electric heating as well fast cooling with the help of liquid nitrogen.

(a)Each dog-bone shaped sample, firstly, was heated to a temperature a few degrees below the glass transition or at glass transition temperature, after fixing the sample in both jaws, the sample was maintained for 5 min in the chamber of the testing machine.(b)The samples were stretched to an elongation equal to 100% at a strain rate of 0.01 s^−1^ and then fast cooled. When the temperature reached 0 °C, the value of tension was measured and then reset.(c)Then, the sample was gradually heated (with the jaws of the handles still closed) with temperature rising rate 2 °C/min and continued observation of the gradual increasing of the stress values, determining on this basis the initial recovery temperature (IRT). The obtained relationship between the temperature change and the value of the observed stress is illustrated in Figure 4, additionally.(d)The other samples, after shaping into a temporary shape using the method described above, were put into the water bath heated to the correct temperature to measure the time of the shape return (tR) and the change of sample length. Since the samples, after giving a temporary shape fastened in the jaws of the testing machine, were quickly cooled and did not show any shrinkage, their fixing ratio was 1, and they also did not show dimensional changes after the period of storage at room temperature; the shape recovery ratio (RR) [37] was calculated according to the following simplified Equation (1).
(1)$$R_{R}\;\%=\frac{\left(e_{m}\;–\;e_{r}\right)}{e_{m}}\times 100$$
where *e_m_*—engineering strain of the temporary shape after unloading, *e_r_*—engineering strain of the recovered permanent shape.

Triple-shape memory properties were determined similarly to the previously described experiment with using Instron 4200 tensile testing:(a)Samples in the dog-bone shape with a gauge length of L_0_ = 35 mm (permanent shape obtained by injection molding at 180 °C were heated in the chamber to 56–57 °C and maintained for 5 min fastened in the jaws;(b)Keeping the temperature around 57 °C, the specimen was stretched at a strain rate 0.01 s^−1^ to about 20% strain, reaching a length of L_1_ = 42 mm; then, it was cooled to 0 °C with stress loaded without opening the jaws;(c)Then, the temperature in the chamber was increased from 0 °C to just below the glass transition temperature T_g_ or T_g2_ of the tested sample, while the next temporary shape with a length of L_2_ = 49 mm was obtained by re-stretching the sample at the same rate and cooled to 0 °C. After removal, the sample was conditioned for one day at room temperature;(d)The next day, the samples obtained previously were introduced to a water bath at a temperature T_g_. The gradual changes of sample length have been noted. When the length of the sample stopped the change, the temperature of the bath was raised up to 57 °C. At this temperature, we noted the start of a further gradual change of the length (results are pictured in Figure 6).

In order to clarify the effect of the triple-shape memory in a more spectacular way, an additional experiment was carried out to demonstrate the three-dimensionality of this phenomenon. For this purpose, the extruded wire (Shape 1) was rolled into the spring shape at a temperature much higher than the glass transition temperature of the used blend (about 70 °C). Thereafter, the obtained shaped sample was quickly cooled, obtaining a temporary Shape 2. Then, this shape was stretched at a temperature close to the glass transition temperature to obtain the second temporary shape (Shape 3). Then, the gradual recovery process from Shape 3 to Shape 2 was tested as a response to repeated heating to the glass transition temperature. After inserting this sample with the recovered Shape 2 into a water bath with a temperature significantly higher than the glass transition temperature, a return to the earlier shape of the wire (Shape 1) was observed. This study is described in detail also in text later and was shown in Figure 5.

## 3. Results and Discussion

The polymer blends were formed by mixing the PLLAGL copolymer with various amounts of BS, BSCA, or CA oligomers (10, 20 and 30 wt %) and extruding in the form of a wire. The glass transition temperatures and mechanical properties of the obtained materials are summarized in Table 2.

The glass transition temperatures given in the table were determined by analyzing the II DSC course (Figure 1).

Neat PLLAGL showed a glass temperature at about 57 °C. The addition of oligoesters lowered the T_g_ of PLLAGL. The glass transition behavior is often used to determine the miscibility of the polymer blend [38]. The fully miscible polymer blends have one single T_g_ that is intermediate between those of the pure components. Two shifted T_g_ values of the blends suggest a molecular chain interpenetrating at the interface, which is considered to be a partially miscible blend. The immiscible polymer blends have two independent T_g_ values, which is the same as with the neat components of the blend.

The blends PLLAGL with 10 and 20 wt % of oligomers BS, BSCA displayed a single glass transition temperature. In the case of blends, PLLAGL with 30 wt % oligoesters BS, BSCA, CA, or 20% CA on DSC thermograms’ endothermic peaks at two different regions were observed. The first peak at a temperature below zero Celsius degrees was attributed to oligomer BS, BSCA, or CA. The second peak at about 40 °C was attributed to PLLAGL. That indicates that blends composed of 70% PLLAGL and 30% oligoesters formed a phase-separated structure (which indicates that separate domains of individual components may be formed, i.e., phase separation occurs). These phenomena indicate that PLLAGL with 30% BS, BSCA blends are compatible but poorly miscible (partially miscible). In the case of blends of PLLAGL with CA, the addition of the oligomer CA in an amount greater than 10% causes a phase separation of produced blends. Table 2 presents T_g_ blend values calculated theoretically based on the Fox equation [39]. According to the blending rule, if the amorphous part of the oligoester blends with the amorphous part of PLLAGL, the glass transition temperature calculated on the basis of the Fox equation should not deviate from the T_g_ determined by thermomechanical techniques. In the case of received materials, the smallest differences in T_g_ values are for PLLAGL blends containing 10% addition of oligoesters. For blends containing 20% oligoesters, differences in T_g_ values are around 10 °C. The reason for such a difference is probably the lack of complete miscibility as well as the occurring hydrogen-bond interactions between the PLLAGL copolymer and the oligomers BS, BSCA, and CA. Similarly, in the work of Deng et al. [40], describing the blends of PLLA and PBS, it was found that the miscibility of these components decreases with increasing PBS content.

Among the used oligomers, the most semi-crystalline is BS. The copolymer PLLAGL is an amorphous material. Among the PLLAGL blends with BS, only the 70/30 blend tends to crystallize. The reason is the presence of crystalline domains associated with an increased amount of BS (Figure 2).

The morphology of the fractured surface of tensile specimens of blends is shown in Figure 3. Obtained micrographs of blends PLLAGL/BS and PLLAGL/BSCA 80/20 confirm coherence between the components. With the composition PLLAGL/BS and PLLAGL/BSCA 70/30, it can be seen in the images that the oligoesters are less well dispersed in PLLAGL. In the blends with CA polyester more than 10 wt % content, small whitish spots can be seen. During the growth of CA content, the phase microseparation is quite evident in the formed blends. It is clearly visible that the blend PLLAGL/CA 70/30 is a kind of phase microseparated system.

These findings clearly suggest that PLLAGL is completely miscible only with BS and BSCA in an amount up to 20%, or with CA in an amount up to 10%. In this case, the results of SEM observations confirmed the DSC data.

Table 2 shows also the mechanical properties of neat PLLAGL and blends PLLAGL/BS, PLLAGL/BSCA, and PLLAGL/CA. Mechanical tests were carried out at room temperature and human body temperature. The mechanical properties were evaluated by measurement of the Young’s modulus and tensile strength. Particularly strong plasticizing properties are presented by oligomers containing citric acid derivatives (CA and BSCA oligomers). The addition of 10% oligoesters to PLLAGL led to a slight decrease in the value of the Young’s modulus at room temperature, but a greater decrease was observed at 37 °C. In addition, in this case, the strength difference was greater when the measurement was conducted at a higher temperature. For the blends PLLAGL/CA 80/20 and 70/30, the value of Young’s modulus and tensile strength decreased dramatically. The main reason for this phenomenon is the highly branched chain structure of a used CA oligomer.

The effects of the addition of oligoesters to the PLLAGL on the shape memory behavior were investigated. Each sample was stretched to 100% after heating to temperature below the glass transition of the blend for 5 min and then cooled below 0 °C. In this way, a temporary shape was received. The sample with temporary shape was put to temperature T_R_ to investigate the shape recovery time (t_R_) and the change of length. The calculated shape recovery ratio (R_R_) and average recovery speed to return to the permanent shape (V_R_) are presented in Table 3.

For all obtained blends, the fastest shape recovery time and greatest shape recovery ratio occur at a temperature close to the glass transition temperature. The lowest values of R_R_ were recorded for PLLAGL with 30% oligoesters. This is related to the limited miscibility of these materials. The highest average recovery speed at the recovery human temperature indicates PLLAGL/BSCA 80/20.

In studies on shape memory behavior, an important parameter is the initial recovery temperature (IRT). In order to determine the IRT, a sample with temporary shape was gradually heated from 0 °C, and the process to return to original shape was observed.

The larger addition of oligoesters in PLLAGL blends significantly decreases the initial recovery temperature. For example, in the case of the PLLAGL copolymer in which the IRT is 45 °C, the addition of 20% of the oligomer BS resulted in a temperature decrease to 36 °C, which is very advantageous from the point of view of medical applications.

For blends obtained with a 10% and 20% addition of BSCA, CA oligoesters, the IRT value speed (V_R_) of the PLLAGL copolymer at 48 °C is 0.03%/s. The addition of oligoesters causes the return speed at a temperature of approximately 48 °C to increase significantly (e.g., the V_R_ of the PLLAGL/BSCA 90/10 blend is 1.24). An important parameter in the study for the shape-memory materials having potential application in the medical field as implants in minimally invasive surgery is the value of the stress generated in the material during return to the programmed shape. The values of the stress depend on the temperature and programming conditions of the temporary shape [3,41,42]. This dependency was confirmed by the conducted tests, the results of which are presented in Figure 4.

The value of the measured stress largely depends on the temperature at which the shape gives and on the type of the tested blend. During shaping at 37 °C (lower than the T_g_ of the all samples), in all cases, the stress value is significantly higher than the stress of the sample deformed at the glass transition temperature, regardless of the measurement temperature. In the case of the PLLAGL copolymer, it is not possible to give a temporary shape by mechanical deformation conducted at 37 °C. The presence of the oligomer in the blend facilitates giving the temporary shape, while the deformation of the sample itself can be carried out at lower temperatures, and it is possible to obtain a higher stress value when the material returns to the programmed shape. The highest stress value (above 6 MPa) is observed for the blend with a 10% addition of the BS oligoester when the deformation to the temporary shape was carried out at 37 °C.

An in-depth test of the shape memory of the obtained blends showed quite unexpectedly that the composites obtained as a result of the mixing in oligo (butylene citrate) or oligo (butylene succinate-*co*-butylene citrate) display an interesting property existence of two separate shape memories. The one temporary shape is associated with the glass transition phase and the second is associated with the intermolecular interactions (we document hereinafter that this is physical cross-linking by hydrogen bonds). The first shape of the material was given by extrusion at 180 °C to obtain the wire (Figure 5-Shape 1). Then, at 70 °C, the second shape (Shape 2) was obtained by rolling the formed wire into a spring. With a turn, the spring was mechanically deformed at a temperature close to T_g_ and then cooled, gaining a shape of stretched spring: Shape 3. The complex shape return process was observed. At a temperature slightly lower than T_g_, in a few minutes, the material returned from Shape 3 to the shape of the spring (Shape 2). When the temperature was raised slightly above Tg, over a minute, the spring was straightened automatically to the wire (Shape 1).

One more test for the dog-bone sample obtained by injection molding based on the blends that contained 20% of oligoesters was performed, too. The first temporary shape of the material was given at a temperature above T_g_ (close to 57–58 °C) by deforming the sample by 20% from an initial length to give the L_1_. Then, a second temporary shape was obtained after lowering the temperature to the T_g_ of the tested material, as a result of another deformation, also amounting to a further 20%, to obtain a length L_2_. The shape recovery process was first tested at a temperature close to T_g_ (T_1_) and then at 57 °C (T_2_). The results of the experiment are shown in the graphs below (Figure 6).

Analyzing the registered dependence of the return rate since the time of experiment (Figure 6), it was possible to observe the existence of two areas, depending on the temperature in which the return to previously programmed shapes took place. The most visible two areas of the return shape are pictured for the blend PLLAGL with 20% of BSCA and a slightly weaker, little noticeable effect for the blend with 20% of CA. For all other blend compositions presented in the work, this effect practically did not occur. For blends with 20% of BSCA or CA, two areas of shape return were observed. The first was at the T_1_ temperature, around 45 °C, and second was at T_2_ above 55 °C. Furthermore, the return to Shape 1 was associated with the phenomenon of phase separation, and glass transition was observed for about 2–3 min for a PLLAGL + 20% BSCA blend or a bit longer for a blend with 20% CA. After this time, the shape change processes practically stopped, and the sample returned to Shape 1 with length L_2_. Until the temperature was raised to T_2_ = 55–57 °C, further transformation was observed. The return in this second temperature area proceeded to Shape 2, the sample of the blend with BSCA after the next 4 min reached the length L_0_ very close to the length of the initial sample before deformation (the degree of a total return of about 98%). In the case of the second blend, the process proceeded lazily, because it lasted about 8–9 min, and the return rate was much smaller. The observed shape return in this temperature area is associated in principle with the intermolecular interactions between the PLLAGL matrix and oligomer CA or BSCA—so, one that has side hydroxyl groups. For this reason, it is most likely that they had to be hydrogen bonds.

In the next stage of the research, attempts were made to confirm the presence and determination of the intermolecular interactions occurring in the obtained blends. The occurrence of such interactions should have a decisive influence, apart from the presence of intermolecular covalent bonds, on the observed shape memory effect. The research was carried out using the analysis of the infrared spectroscopy. FTIR spectra recorded for blends of PLLAGL with BS, CA, and BSCA were compared with the spectra of particular components.

The most interesting bands from the studied point of view appeared in the region of 1750–1650 cm^−1^ corresponding to stretching vibrations of C=O ester groups. In the case of the BS oligomer in this region, the band with two clearly visible maxima at 1720 and 1712 cm^−1^ that originated from vibrations in intermediate and crystalline phase [41] were observed. In addition, some low-intensity shoulders were seen; the second derivatives spectra indicated that the bands at 1738 and 1731 cm^−1^ were assigned to vibrations in the amorphous phase. In the case of the CA oligomer, the second derivative spectra demonstrated the presence of four bands: at 1739, 1728, 1724, and 1711 cm^−1^ assigned to stretching vibrations of free and bonded C=O ester and acid groups, respectively [34]. The bands attributed to the stretching vibrations of OH groups appeared at about 3480 cm^−1^, thus in the region due to bonded OH groups. The BSCA oligomer included both BS and CA units, which was reflected in the FTIR spectra; however, the relations between particular bands were different. If we compare the spectrum of the BSCA oligomer with the spectrum being the sum of the CA and BS spectra (Figure 7), it can be seen that in the BSCA spectrum, the band at 1712 cm^−1^ was relatively higher, similar to that at 1728 cm^−1^. This may suggest that in the case of the BSCA oligomer, there are more free C=O ester groups that originated from CA oligomer (1728 cm^−1^). Due to the lack of bands characteristic for free OH groups in the spectrum of BSCA, it should be assumed that the OH groups formed hydrogen bonds with the C=O ester groups from the BS oligomer (1712 cm^−1^).

FTIR 2D spectra of PLLAGL indicated in the studied region the band at 1746 cm^−1^ with a shoulder at about 1753 cm^−1^, which according to the literature [43] corresponds to C=O ester groups vibrations in the amorphous and crystalline phase, respectively. To characterize changes in FTIR spectra after blending PLLAGL with oligomers, the blends with higher contents (30%) of oligomers were used. In the PLLAGL/CA blend, only a broad band with a maximum at 1750 cm^−1^ was observed (Figure 8). After blending PLLAGL with oligomers, the intensity of the band at 1753 cm^−1^ increased, reflecting the increase in the crystallinity of this unit. It is particularly visible for the PLLAGL/BS blend, in which no band at 1746 cm^−1^ was shown by 2D spectra. In the case of PLLAGL/BS and PLLAGL/BSCA, the bands that originated from oligomers in that region changed a little in contrast to PLLAGL/CA, for which 2D spectra indicated one broad band at about 1728 cm^−1^ assigned to free C=O ester groups. Considering that the band derived from OH groups has not changed, it should be concluded that in that case, OH groups from the CA oligomer interacted with C=O groups from PLLAGL, forming hydrogen bonds. Thus, blending has caused an emergence of interactions between PLLAGL and CA, while in the other two cases, it caused changes in the ordering of particular blend components. In the case of PLLAGL/BSCA, the hydrogen bonds between the BS and CA components have been preserved, as it was in the BSCA oligomer.

We admit that these differences in interactions in the blends can be a reason for the occurrence of a second shape memory for PLLAGL/BSCA blend. Consequently, for the studied blends, the FTIR spectra at the temperatures of shape memory, i.e., at 37 °C and 57 °C, were recorded.

As seen in Figure 9, only slight changes in the analyzed region were detected in the cases of PLLAGL/BS and PLLAGL/CA.

The situation was different for the blend with 20% BSCA. In that case, at 57 °C, the intensities of the bands at 1712 and 1721 cm^−1^ that originated from the stretching vibrations of bonded C=O groups in CA and BS components decreased significantly, while the intensity of the bands at 1730 cm^−1^ due to the vibrations of free C=O groups notably increased, which indicates the breaking of hydrogen bonds at this temperature. Thus, the observed second shape memory most probably is a result of the presence the hydrogen bonds in the PLLAGL/BSCA blend.

## 4. Conclusions

The addition of biocompatible oligomers obtained by the condensation of succinic and citric acids with butanediol BSCA to the PLLAGL copolymer contributes to changes not only on the glass transition temperature and the final material stiffness, but it also modifies the shape memory behavior of the resulted blend. The presence of the oligoesters in the blend makes it easier to obtain a temporary shape due to the plasticizing effect. Deformation to the temporary shape of the sample can be carried out at temperatures significantly lower than the glass transition temperature of the blend. It becomes possible to achieve high stresses during self-returning to a permanent shape, which can be particularly useful in applications of this material in the formation of tools performing mechanical work as a response to the temperature change.

Among the tested blends, PLLAGL/BS and PLLAGL/BSCA materials composed of 90/10 and 80/20 look to be especially promising in biomedical applications. These blends exhibit a compatibility of components and present one glass transition temperature. The addition of 30% BS or 20% CA causes the blend obtained to not be fully compatible. These materials present two glass transition temperatures, where the lower one is related to the phase of the introduced oligoester, and the higher one is related to the PLLAGL copolymer phase. A similar phenomenon was observed by Park [31] in studies of a similar PLLA blend, where a clear phase separation was observed in the polarizing microscope image.

The blends of PLLAGL with 20 wt % of the oligoesters containing hydroxyl side groups, oligo (butylene citrate) or oligo (butylene succinate-co-butylene citrate), present triple shape memory behavior. This effect is especially visible for the compatible blends with 20% of oligo (butylene succinate-*co*-butylene citrate). The memory of the first shape is mainly related to the presence of intermolecular interactions, the formation of hydrogen bonds between the side OH groups of the oligomer, and the oxygen atoms of the PLLAGL copolyester chain. The memory of the second shape is the effect of the occurrence of phenomenon of phase separation and entanglement of the polymer chains. Thus, the presented memory of many shapes for this type of blend occurs only when the blends are compatible and contain a sufficiently high content of oligomers with side hydroxyl groups.

It is possible to design a polymer or a blend of polymers to obtain different types of intermolecular interactions in the same formed material, so that each of them could cause a possibility to return to the previously programmed shape, which is stimulated by different temperatures.

The selected blend, which is relatively easy to obtain by mixing and extrusion, has many advantages compared to the numerous previously described shape memory materials intended for use in medicine. It is a biocompatible and biodegradable material without additives commonly used during blending. Only PLLAGL with proven biocompatibility and aliphatic oligoesters obtained from esters of succinic and citric acids (the acids of Krebs cycle) form the blend composition. The obtained blend is a thermoplastic, with a homogeneous structure, that is easily processable, which we confirmed in the conducted research on forming vascular stents by microinjection molding. The temperatures of return are close to the body temperature. Most of the described previously blends with shape memory properties so far are immiscible and therefore difficult to process, and most often, they have weak strength. The presented blend shows good mechanical properties; the tensile strength is slightly lower than that of PLLAGL and more flexible. What is important is that, using the blend, it is possible to obtain a relatively good recovery stress (in the range of 3–4 MPa). This is an important feature needed in the case of forming various actuators (self-expanding vascular stents, occlusion tools, etc.).

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
