# Peer review of "Triple-Shape Memory Behavior of Modified Lactide/Glycolide Copolymers"

_polymers, 2020, doi:10.3390/polym12122984_

Round 1
Reviewer 1 Report
In this manuscript 27 of 28 references are dated before 2015. The more recent publications are not mentioned and not discussed. Does it mean that in the last 5 years no essential contribution has been made in the state-of-the-art? In my opinion the relevant contributions into the state-of-art in time period 2016 - 2020 must be included in Introduction as well, because the lack of this information don't allow to formulate clearly the novelty in Conclusions.
The biomedical tests of developed materials have not been included in the research design so a specific biomedical application has not been clearly presented in conclusions and advantages over currently suggested similar biomaterials have not been demonstrated and competitiveness of developed compositions also has not been demonstrated. However experimental data presented in the current version of manuscript could be useful for researchers in the area of polymer blends
Author Response
Thank you for your opinion. We agree with your suggestions. In the revised version, we tried to introduce the all recommended changes.
In this manuscript 27 of 28 references are dated before 2015. The more recent publications are not mentioned and not discussed. Does it mean that in the last 5 years no essential contribution has been made in the state-of-the-art? In my opinion the relevant contributions into the state-of-art in time period 2016 - 2020 must be included in Introduction as well, because the lack of this information don't allow to formulate clearly the novelty in Conclusions.
In line with your apt suggestion, we introduced descriptions of newer achievements in this presented field. The newly entered text, as well as the corrected one, is marked in red.
The biomedical tests of developed materials have not been included in the research design so a specific biomedical application has not been clearly presented in conclusions and advantages over currently suggested similar biomaterials have not been demonstrated and competitiveness of developed compositions also has not been demonstrated. However experimental data presented in the current version of manuscript could be useful for researchers in the area of polymer blends
In response to your comment, in the conclusions section, we introduced a relevant comments related to this topic.
The described shape memory polymeric blend appears to be useful for most of the previously proposed in literature different biomedical applications. In our research, we use this material successfully in the formation of vascular stents showing the property of self-expansion after implantation. Of course, the results of this research will be the subject of the completely separate publications.
The selected blend, relatively easy to obtain by mixing and extrusion, has many advantages compared to numerous previously described shape memory materials intended to use in medicine. It is a biocompatible and biodegradable material, without additives commonly used during blending. Only PLLAGA with proven biocompatibility and aliphatic polyesters obtained from esters of succinic and citric acids (the acids of Krebs cycle) form the blend composition. The obtained blend is a thermoplastic, with a homogeneous structure, easily processable, which we confirmed in the conducted research on forming vascular stents by microinjection moulding. The temperatures of return are close to the body temperature. Most of the described blends with shape memory properties so far are immiscible, most often with weak strength. The presented blend shows good mechanical properties, the tensile strength is slightly lower than that of PLAGA, more flexible. What is important, using the blend, is possible to obtain relatively good recovery stress (in the range of 3 - 4 MPa). This is an important feature needed in the case of forming various actuators (self-expanding vascular stents, occlusion tools etc.).
We are using the triple - shape memory property of the described blends during test production of self-expanded stents by micro-injection moulding for thin-walled, complex-shaped products, products that cannot be practically removed from the mould after it has cooled completely. The memory property of the three shapes allows remove product from the mould with the designed shape, even in the case of deformation (temporary shape obtained at a temperature above the glass transition temperature). We have included a large part of the above comment in Conclusion part of the manuscript.
Reviewer 2 Report
Revision is required.
- Extensive language editing is required.
- As reported in
Thermo/chemo-responsive shape memory effect in polymers: a sketch of working mechanisms, fundamentals and optimization, Journal of Polymer Research, Vol. 19, No. 9, 2012, 9952,
most polymers are heat/chemo-responsive SMP. There are a couple of working mechanism for the SME in SMPs, cross-linking is not a must.
- The triple-SME is an intrinsic feature of most SMPs, as explained in
Mechanisms of the multi-shape memory effect and temperature memory effect in shape memory polymers, Soft Matter, Vol. 6, 2010, pp4403-4406
“…a broad transition temperature range” is not a must.
- “It is well known that PLLA is a hard and brittle material at room temperature, therefore it is not possible to deform it to 100% without destroying the sample. In order to improve the PLLA's ability to plastic deformation,..”
Pls explain when we need high plastic deformation at room temperature in PLLA based biomedical devices.
- “at room temperature”
Pls name the actual room temperature at beginning.
- “ΔH – heat of melting of crystalline phase”
Although the latent heat in both melting and crystallization of a material is a constant, dur to energy dissipation nature in any transition, ΔH in melting and crystallization is not a constant.
- A heating rate of 20°C/min in DSC is a bit too high. A speed of 10 oC/min is more appropriate.
- Pls provide the thickness of dog-bone shaped samples.
- “The rate of stretching was 20 mm/min”
This is sample size dependent loading speed. Pls provide strain rate as a size independent loading speed.
- “testing:(a) each sample was first heated to temperature below the glass transition of the material and maintained for 5min; (b) then the samples were stretched using a testing machine to an elongation equal to 100% (em) (or less than 100% when the sample was destroyed) at a rate of 20 mm/min. at a temperature above the glass transition of the material, and the cooled. When the temperature reached 0°C, the tension was measured; ”
Pls double check steps (a) and (b) wrt temperature. How do u manage to cool to 0°C?
- “(c) sample was then gradually heated with temperature rising rate 2°C/min and continued observation of the process of return to its permanent shape by measuring the initial recovery temperature (IRT)”
Clamps should be removed first. If the shape of the pre-stretched sample is fixed, this is called constrained recovery. The sample should not be able to return its original shape. We use this approach to check the recovery stress (refer to above
Mechanisms of the multi-shape memory effect and temperature memory effect in shape memory polymers). For the discussion wrt Fig 4, pls refer to this reference as well to reword about the influence of programming temperature.
- 1, the definition of the shape recovery ratio is wrong.
Pls refer to
Characterization of polymeric shape memory materials, Journal of Polymer Engineering, Vol. 37, 2017, 1-20
for the right one.
Shape fixity ratio is missing.
- “(a) samples with a length of L0 (permanent shape) was heated to temperature 57°C and maintained for 5min; (b) specimen was stretched at a stress rate 20mm/min to about 20% strain reaching a length of L1, then it was cooled to 0°C with stress loaded; (c) the next temporary shape with a length of L2 obtained by re-stretching at the same rate to about 20% strain (to the first temporary shape) and cooled to 0°C; (d) the process of returning to the original shape was observed at glass temperature of material and at 57°C, respectively.”
Pls carefully check above steps.
Gauge length is not provided; stress rate should be strain rate or stretching rate; cooled to 0 oC?; in (c), does another 20% strain refer to the original gauge length?
Step (d) to check the triple SME: need to check if the shape upon heating to the glass transition temperature is stable. Hence, need to monitor the continuous shape change in the heating process.
- Fig 5. T1 to T2 is confusion. Does not match the description in the caption. Fig 5 is not seemingly mentioned in the experimental part.
- Fig 6 should be against temperature, since buckling (or curling) may happen due to temperature gradient in the pre-stretched sample upon heating. (refer to
Instability/collapse of polymeric materials and their structures in stimulus-induced shape/surface morphology switching, Materials and Design, Vol. 59, 2014, 174-192)
Slow gradual heating is preferred instead of fixing a high temperature.
Author Response
Revision is required.
- Extensive language editing is required.
We have improved the grammar and syntax errors of the text
- As reported in Thermo /chemo-responsive shape memory effect in polymers: a sketch of working mechanisms, fundamentals and optimization, Journal of Polymer Research, Vol. 19, No. 9, 2012, 9952, most polymers are heat/chemo-responsive SMP. There are a couple of working mechanism for the SME in SMPs, cross-linking is not a must.
Yes, we agree with that (see line 44-50 in the manuscript text). In our case, the one temporary shape is associated with the glass transition phase and the second is associated with the intermolecular interactions (we document with FTIR tests, that this is probably mainly physical cross-linking by hydrogen bonds of the oligomers chains contained units of butylene citrate - side OH groups).
- The triple-SME is an intrinsic feature of most SMPs, as explained in Mechanisms of the multi-shape memory effect and temperature memory effect in shape memory polymers, Soft Matter, Vol. 6, 2010, pp4403-4406
“…a broad transition temperature range” is not a must.
Yes, we agree with that (see line 55-59 and 70-72 in the manuscript text).
- “It is well known that PLLA is a hard and brittle material at room temperature, therefore it is not possible to deform it to 100% without destroying the sample. In order to improve the PLLA's ability to plastic deformation,..”
Pls explain when we need high plastic deformation at room temperature in PLLA based biomedical devices.
This is our mistake that occurred while editing the manuscript. Of course, we do not program the temporary shape at room temperature but at 37 C (body temperature), the temperature below to the glass transition temperature of the most tested blends. Then we get the highest recovery stress (see fig. 4).
It is known that programing the temporary shape of the products formed with PLLA by significant mechanical deformation at temperature close to glass transition, is practically impossible without destroying the sample. In this case, the highly deformed temporary shape can only be formed at temperatures higher than the glass transition temperature. These reasons that the process of the return to permanent shape proceed with a relatively low recovery stress (recovery force). [Janice J. Song, Huntley H. Chang, Hani E. Naguib, Biocompatible shape memory polymer actuators with high force capabilities, European Polymer Journal, Volume 67, 2015, Pages 186-198].
Pls name the actual room temperature at beginning.
It is usually taken between 20 °C and 25 °C, but after the amendments in text it doesn't matter anymore.
- “ΔH – heat of melting of crystalline phase”
Although the latent heat in both melting and crystallization of a material is a constant, dur to energy dissipation nature in any transition, ΔH in melting and crystallization is not a constant.
We agree that the value of this measured heat is largely dependent on the measurement conditions and the type of DSC transition. Therefore, based on your attention, we have introduced brief information on the conditions of the DSC measurement (which run, speed of temperature change). The heats of change determined in this way serve rather to demonstrate the presence of semicrystalline phases in the tested material and the possibility of a relative comparison of the degree of semicrystallinity between different oligomers.
- A heating rate of 20°C/min in DSC is a bit too high. A speed of 10 °C/min is more appropriate.
You are right, in many cases, especially in the measurement of thermal effects related to crystallization and melting, a slower rate of heat change, around 10 °C / min when performing a DSC measurement, is preferable. However, based on our experience in testing this type of materials, higher speed is on the other hand safer, as it limits the effects of thermal degradation that may occur. In our case, the task was mainly to determine the glass transition temperature of the tested materials. The measurement was carried out according to the ASTM E 1356-08 standard (Standard Test Method for Assignment of the Glass Transition Temperatures by Differential Scanning Calorimetry), according to a standard that has just been prepared for this type of measurement. We have introduced a reference to the standard number in the text.
- Pls provide the thickness of dog-bone shaped samples.
We introduced the missing description of the conducted tensile strength test, where we described the dimensions of the tested samples.
- “The rate of stretching was 20 mm/min”. This is sample size dependent loading speed. Pls provide strain rate as a size independent loading speed.
You are absolutely right, we have changed the used wrong term – was introduced “strain rate” term.
- “testing:(a) each sample was first heated to temperature below the glass transition of the material and maintained for 5min; (b) then the samples were stretched using a testing machine to an elongation equal to 100% (em) (or less than 100% when the sample was destroyed) at a rate of 20 mm/min. at a temperature above the glass transition of the material, and the cooled. When the temperature reached 0°C, the tension was measured; ”
Pls double check steps (a) and (b) wrt temperature. How do u manage to cool to 0°C?
We have edited this part of the text taking into account your comments.
- “(c) sample was then gradually heated with temperature rising rate 2°C/min and continued observation of the process of return to its permanent shape by measuring the initial recovery temperature (IRT)”
Clamps should be removed first. If the shape of the pre-stretched sample is fixed, this is called constrained recovery. The sample should not be able to return its original shape. We use this approach to check the recovery stress (refer to above Mechanisms of the multi-shape memory effect and temperature memory effect in shape memory polymers). For the discussion wrt Fig 4, pls refer to this reference as well to reword about the influence of programming temperature.
We have made changes to the description of this study to make it comprehensive and understandable.
1, the definition of the shape recovery ratio is wrong.
Pls refer to Characterization of polymeric shape memory materials, Journal of Polymer Engineering, Vol. 37, 2017, 1-20 for the right one.
We rewrite the equation ; according to: United States Pat., US 2013/0018111A1, 2013. A. Lendlein and S. Kelch, Angew. Chem., Int. Ed., 2002, 41, 24. An appropriate comment has been entered.
Shape fixity ratio is missing.
Yes, because in the conducted test the programming of the temporary shape was carried out in such a way that the shrinkage effect did not occur (sample cooling under load below the glass transition temperature). Therefore fixing ratio was practically 1. We have introduced the appropriate explanation in the text.
- “(a) samples with a length of L0 (permanent shape) was heated to temperature 57°C and maintained for 5min; (b) specimen was stretched at a stress rate 20mm/min to about 20% strain reaching a length of L1, then it was cooled to 0°C with stress loaded; (c) the next temporary shape with a length of L2 obtained by re-stretching at the same rate to about 20% strain (to the first temporary shape) and cooled to 0°C; (d) the process of returning to the original shape was observed at glass temperature of material and at 57°C, respectively.”
Pls carefully check above steps.
Gauge length is not provided; stress rate should be strain rate or stretching rate; cooled to 0 oC?; in (c), does another 20% strain refer to the original gauge length?
Step (d) to check the triple SME: need to check if the shape upon heating to the glass transition temperature is stable. Hence, need to monitor the continuous shape change in the heating process.
We have thoroughly edited the text of the description of the stage of testing the memory of two shapes carried out in the testing machine chamber.
- Fig 5. T1 to T2 is confusion. Does not match the description in the caption. Fig 5 is not seemingly mentioned in the experimental part.
We changed the description as recommended. We added a short description of the test in the experimental parts too.
- Fig 6 should be against temperature, since buckling (or curling) may happen due to temperature gradient in the pre-stretched sample upon heating. (refer to Instability/collapse of polymeric materials and their structures in stimulus-induced shape/surface morphology switching, Materials and Design, Vol. 59, 2014, 174-192)
Slow gradual heating is preferred instead of fixing a high temperature.
We are agreeing with your recommendation, but in our case the speed of the shape return was too fast. Our intention was measurement rather speed of returning to the both shapes, not strict analysis of the process of shape transition.
Round 2
Reviewer 2 Report
Minor revision is required.
- “The shape memory effect of several shapes is explained by the 57 occurrence of a few glass transition temperature or it broad range, as well the occurrence of two or 58 more separate molecular switches related to the structure of the polymer chain (discrete thermal 59 transitions, two independent switch units associated with two different glass transitions 60 temperatures, molecular switches by covalent bonds and supramolecular interactions either by 61 themselves or in combination with a phase transition switch) [6-9].”
Still “broad range”.…occurrence of a few glass transition temperature or it broad range
It is not necessary to be broad range, and it is not limited to glass transition as well. Refer to Mechanisms of the multi-shape memory effect and temperature memory effect in shape memory polymers, Soft Matter, Vol. 6, 2010, pp4403-4406 for explanation.
- “…temperature of the most tested blends. Then we get the highest recovery stress (see fig. 4)…. These reasons that the process of the return to permanent shape proceed with a relatively low recovery stress (recovery force).”
This has been explained in Optimization of the shape memory effect in shape memory polymers, Journal of Polymer Science Part A: Polymer Chemistry, Vol. 46, No. 16, 2011, pp3582-3587
Also refer to Figure 31 in
- Wu, X.; Huang, W.M.; Zhao, Y.; Ding, Z.; Tang, C.; Zhang, J. Mechanisms of the Shape Memory Effect 576 in Polymeric Materials. Polymers 2013, 5, 1169-1202.
for an example with PMMA.
- “strain rate of 20 mm/min”
This is a sample size dependent speed. The unit of strain rate is /s. It can be worked out via speed (in mm/min)/gauge length…
- “because in the conducted test the programming of the temporary shape was carried out in such a way that the shrinkage effect did not occur (sample cooling under load below the glass transition temperature). Therefore fixing ratio was practically 1”
This procedure does not guarantee the shape fixity ratio is 1. The shape fixity ratio depends on how soft a polymer is during stretching, and subsequent creeping/relaxation, which is commonly observed in many polymers.
- The selected blend, relatively easy to obtain by mixing and extrusion, has many advantages 551 compared to numerous previously described shape memory materials intended to use in medicine.
The problem in applying PLA in biomedical devices (e.g., self-expanding vascular stents, occlusion 563 tools etc. as mentioned at the end this paper) is its high activation temperature. This is the same (and even worse) in the materials developed here as well. One of the most recent developments is to apply water-activated shape recovery, which does not need heating to above body temperature at all.
Author Response
Thank you for the thorough analysis of our work and valuable comments with which we mostly agree.
Minor revision is required.
- “The shape memory effect of several shapes is explained by the 57 occurrence of a few glass transition temperature or it broad range, as well the occurrence of two or 58 more separate molecular switches related to the structure of the polymer chain (discrete thermal 59 transitions, two independent switch units associated with two different glass transitions 60 temperatures, molecular switches by covalent bonds and supramolecular interactions either by 61 themselves or in combination with a phase transition switch) [6-9].”
Still “broad range”.…occurrence of a few glass transition temperature or it broad range
It is not necessary to be broad range, and it is not limited to glass transition as well. Refer to Mechanisms of the multi-shape memory effect and temperature memory effect in shape memory polymers, Soft Matter, Vol. 6, 2010, pp4403-4406 for explanation.
We agree that the shape memory of a polymer can exist independently of the glass transition temperature, and is not limited to its existence. In the introduction part, we write about it, citing many works. According to your suggestion, we have removed the unfortunate discussed phrase "broad range". We also added the publication “Mechanisms of the multi-shape memory effect and temperature memory effect in shape memory polymers, Soft Matter, Vol. 6, 2010, pp4403-4406.” to the list of citations with short comment.
- “…temperature of the most tested blends. Then we get the highest recovery stress (see fig. 4)…. These reasons that the process of the return to permanent shape proceed with a relatively low recovery stress (recovery force).”
This has been explained in Optimization of the shape memory effect in shape memory polymers, Journal of Polymer Science Part A: Polymer Chemistry, Vol. 46, No. 16, 2011, pp3582-3587
Also refer to Figure 31 in
- Wu, X.; Huang, W.M.; Zhao, Y.; Ding, Z.; Tang, C.; Zhang, J. Mechanisms of the Shape Memory Effect 576 in Polymeric Materials. Polymers 2013, 5, 1169-1202.
for an example with PMMA.
We supplemented the text with suggested citation .
- “strain rate of 20 mm/min”
This is a sample size dependent speed. The unit of strain rate is /s. It can be worked out via speed (in mm/min)/gauge length…
Thank you especially for this attention. We have corrected this error by recalculating as prompted (strain rate = cross head speed/gauge length).
cross head speed 20 mm/min ≈ 0.333 mm/s strain rate = 0.333 [mm/s] /35 [mm] ≈ 0.01 s-1
- “because in the conducted test the programming of the temporary shape was carried out in such a way that the shrinkage effect did not occur (sample cooling under load below the glass transition temperature). Therefore fixing ratio was practically 1”
This procedure does not guarantee the shape fixity ratio is 1. The shape fixity ratio depends on how soft a polymer is during stretching, and subsequent creeping/relaxation, which is commonly observed in many polymers.
This is true, but in our case the dimension remained unchanged, because the fixing sample was quickly frozen after being stretched without first removing the machine clamps. Therefore we assume that the fixing ratio was equal to 1. It may be an approximation, but it does not significantly affect the final calculated results.
- The selected blend, relatively easy to obtain by mixing and extrusion, has many advantages 551 compared to numerous previously described shape memory materials intended to use in medicine.
The problem in applying PLA in biomedical devices (e.g., self-expanding vascular stents, occlusion 563 tools etc. as mentioned at the end this paper) is its high activation temperature. This is the same (and even worse) in the materials developed here as well. One of the most recent developments is to apply water-activated shape recovery, which does not need heating to above body temperature at all.
I agree with you on this completely. Providing heat, in our case especially above the glass transition temperature, is very difficult and complicates the construction of the actuator. On the other hand, it seems very difficult to obtain significant recovery forces when using the materials that respond to stimuli other than temperature.